ecology, molecular biology, microbiology

amplicon sequencing, bacteria, co-occurrence, diet, fungi, host–microbe interactions

**Authors for correspondence:**
Xavier A. Harrison
e-mail: x.harrison@exeter.ac.uk
Rachael E. Antwis
e-mail: rachael.antwis@gmail.com

# Fungal microbiomes are determined by host phylogeny and exhibit widespread associations with the bacterial microbiome

Xavier A. Harrison[1], Allan D. McDevitt[2], Jenny C. Dunn[3], Sarah M. Griffiths[4], Chiara Benvenuto[2], Richard Birtles[2], Jean P. Boubli[2], Kevin Bown[2], Calum Bridson[4,5], Darren R. Brooks[2], Samuel S. Browett[2], Ruth F. Carden[6,7], Julian Chantrey[8], Friederike Clever[4,9], Ilaria Coscia[2], Katie L. Edwards[10], Natalie Ferry[2], Ian Goodhead[2], Andrew Highlands[2], Jane Hopper[11], Joseph Jackson[2], Robert Jehle[2], Mariane da Cruz Kaizer[2], Tony King[11,12], Jessica M. D. Lea[5], Jessica L. Lenka[2], Alexandra McCubbin[13], Jack McKenzie[2], Bárbara Lins Caldas de Moraes[14], Denise B. O'Meara[15], Poppy Pescod[2], Richard F. Preziosi[4], Jennifer K. Rowntree[4], Susanne Shultz[5], Matthew J. Silk[1], Jennifer E. Stockdale[13,16], William O. C. Symondson[13], Mariana Villalba de la Pena[5], Susan L. Walker[10], Michael D. Wood[2] and Rachael E. Antwis[2]

[1]School of Biosciences, University of Exeter, UK
[2]School of Science, Engineering and Environment, University of Salford, UK
[3]School of Life Sciences, Joseph Banks Laboratories, University of Lincoln, UK
[4]Ecology and Environment Research Centre, Department of Natural Sciences, Manchester Metropolitan University, UK
[5]Department of Earth and Environmental Sciences, University of Manchester, UK
[6]School of Archaeology, University College Dublin, Ireland
[7]Wildlife Ecological and Osteological Consultancy, Wicklow, Ireland
[8]Institute of Veterinary Science, University of Liverpool, UK
[9]Smithsonian Tropical Research Institute, Ancon, Republic of Panama
[10]North of England Zoological Society, Chester Zoo, Upton-by-Chester, UK
[11]The Aspinall Foundation, Port Lympne Reserve, Hythe, Kent, UK
[12]School of Anthropology and Conservation, University of Kent, UK
[13]School of Biosciences, University of Cardiff, UK
[14]Department of Zoology, Federal University of Pernambuco, Recife, Brazil
[15]School of Science and Computing, Waterford Institute of Technology, Ireland
[16]School of Life Sciences, University of Nottingham, UK

XAH, 0000-0002-2004-3601; ADM, 0000-0002-2677-7833; JCD, 0000-0002-6277-2781; SMG, 0000-0003-4743-049X; CB, 0000-0002-8378-8168; RB, 0000-0002-4216-5044; JPB, 0000-0002-5833-9264; SSB, 0000-0003-3469-9609; RFC, 0000-0002-2829-4667; JC, 0000-0002-4801-7034; IC, 0000-0001-5768-4675; JJ, 0000-0003-0330-5478; RJ, 0000-0003-0545-5664; TK, 0000-0003-4597-0134; BLCdM, 0000-0002-1804-9828; DBOM, 0000-0003-1372-5505; SS, 0000-0002-7135-4880; WOCS, 0000-0002-3343-4679; SLW, 0000-0003-3187-5195; REA, 0000-0002-8849-8194

Interactions between hosts and their resident microbial communities are a fundamental component of fitness for both agents. Though recent research has highlighted the importance of interactions between animals and their bacterial communities, comparative evidence for fungi is lacking, especially in natural populations. Using data from 49 species, we present novel evidence of strong covariation between fungal and bacterial communities across the host phylogeny, indicative of recruitment by hosts for specific suites of microbes. Using co-occurrence networks, we demonstrate marked

variation across host taxonomy in patterns of covariation between bacterial and fungal abundances. Host phylogeny drives differences in the overall richness of bacterial and fungal communities, but the effect of diet on richness was only evident in the mammalian gut microbiome. Sample type, tissue storage and DNA extraction method also affected bacterial and fungal community composition, and future studies would benefit from standardized approaches to sample processing. Collectively these data indicate fungal microbiomes may play a key role in host fitness and suggest an urgent need to study multiple agents of the animal microbiome to accurately determine the strength and ecological significance of host–microbe interactions.

## 1. Introduction

Multicellular organisms support diverse microbial communities critical for physiological functioning, immunity, development, evolution and behaviour [1–3]. Variability in host-associated microbiome composition may explain asymmetries among hosts in key traits including susceptibility to disease [4,5], fecundity [6] and resilience to environmental change [7]. Although the microbiota is a complex assemblage of bacteria, fungi, archaea, viruses and protozoa, the overwhelming majority of research has focused solely on the bacterial component [8,9]. Although relatively well documented in soils and plants [10–13], relatively few studies have examined the dynamics of non-bacterial components of the microbiome in animal hosts (but see [14–16]), especially in non-model or wild systems. As such, our current understanding of host–microbe interactions is skewed by a bacteria-centric view of the microbiome. Although not well understood, there is growing evidence that the fungal microbiota, termed the 'mycobiome', may drive diverse functions such as fat, carbon and nitrogen metabolism [17,18], degradation of cellulose and other carbohydrates [19], pathogen resistance [20], initiation of immune pathways and regulation of inflammatory responses [9,21], and even host dispersal [22].

Host phylogeny has repeatedly been shown to be an important predictor of bacterial microbiome structure in multiple vertebrate clades, a phenomenon known as 'phylosymbiosis' [23–27]. This phenomenon often reflects phylogenetic patterns in life-history traits, such as diet, physiology or spatial distribution [23–27]. However, evidence of phylosymbiosis, and its drivers, in other microbial kingdoms or domains is lacking. Addressing this major gap in our knowledge is crucial as we likely underestimate the strength and importance of coevolution between animal hosts and their resident communities, particularly in the context of cross-kingdom interactions within the microbiome [28].

Here, we used ITS and 16S rRNA gene amplicon sequencing to characterize fungal and bacterial communities of primarily gut and faecal samples from 49 host species across eight classes, including both vertebrates and invertebrates (electronic supplementary material, table S1). We predicted that both fungal and bacterial microbiomes would demonstrate strong signals of phylosymbiosis across the broad host taxonomic range tested. Specifically, we predicted that patterns of phylosymbiosis within microbial kingdoms will also drive significant positive covariance in patterns of microbial community structure between microbial kingdoms within individual hosts, suggestive of evolutionary

constraints that favour co-selection of specific bacterial and fungal communities in tandem. We also used network analysis to identify key bacteria–fungi associations while quantifying variation in the composition and structure of bacteria–fungi networks across host taxonomic groups. Finally, we tested the prediction that cross-kingdom phylosymbiosis may be partially driven by similarity in host dietary niche across the 32 bird and mammal species sampled.

## 2. Results

### (a) Fungal and bacterial microbiome diversity varies with host phylogeny

Our data revealed consistent patterns in fungal and bacterial alpha-diversity across host taxonomic groups. Bacterial community alpha-diversity was generally greater than, or similar to, fungal community alpha-diversity at the host species level (figure 1a), although two species exhibited greater fungal diversity than bacterial (great tit, tsetse fly; figure 1a). Comparisons between microbial richness values within individuals (i.e. *relative* richness) using a binomial GLMM supported these patterns, indicating that bacterial richness was higher on average than fungal in 80% of individuals [95% credible interval 0.43–0.96]. When conditioning on class, samples from both Mammalia and Insecta were more likely to have higher bacterial diversity than fungal diversity (credible intervals not crossing zero on the link scale; electronic supplementary material, figure S2). Mammalia were 32.7% more likely to have higher bacterial relative to fungal diversity than Aves in our study organisms (95% credible interval 7.6–58.7%). Variation among species in this model explained 28.8% (16.1–39.7%) of the variation in relative microbial richness. Using a bivariate model with both fungal and bacterial diversity as response variables to examine patterns of absolute microbial richness across host taxonomy, both Mammalia and Insecta exhibited bacterial diversity that was consistently greater than fungal diversity when controlling for variation among species (credible intervals of mean difference between diversity estimates not crossing zero). There was no evidence of positive covariance between fungal and bacterial richness values at the species level (mean correlation 0.18, 95% credible intervals −0.34 to 0.65), suggesting that the high diversity of one microbial group does not necessarily reflect high diversity of the other. The bivariate model, containing effects of class and species, explained 53.3% (44.7–60.3%) of variation in bacterial Shannon diversity and 29.8% (16.3–40.8%) of variation in fungal Shannon diversity.

Phylogenetic analyses supported these general patterns (electronic supplementary material, figure S3). We detected phylogenetic signal in observed amplicon sequence variants (ASVs) for both fungi ($C_{mean} = 0.27$, $p_{BH} = 0.004$) and bacteria ($C_{mean} = 0.29$, $p_{BH} = 0.004$). For Shannon diversity, no significant relationship was present for fungi ($p_{BH} = 0.079$) or bacteria ($p_{BH} = 0.062$) after correction for multiple testing.

### (b) Limited evidence of covariation between host diet and fungal microbiome

#### (i) Alpha-diversity

Models exploring the influence of diet on microbial richness yielded mixed results. In mammals, we detected a relationship

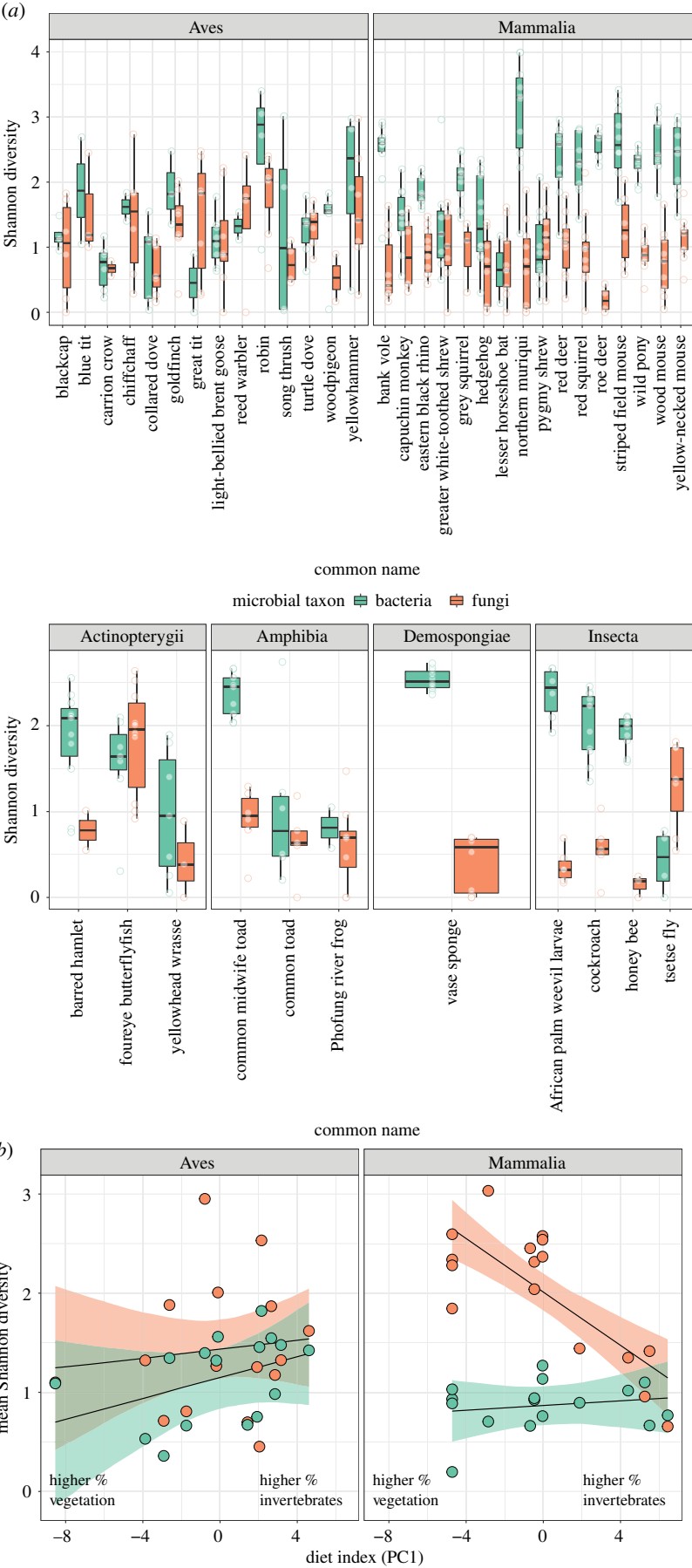

**Figure 1.** Host phylogeny and diet as predictors of host bacterial and fungal alpha-diversity. (a) Boxplots and raw data (points) of inverse Simpson indices for bacterial (green) and fungal (orange) communities across a range of host species. Note different y-axis scales for the two rows. (b) Raw data (points) and model predictions (shaded area and lines) of models examining the relationship between host diet and microbiome alpha-diversity. In mammals, an increase in the amount of plant material in the diet (more negative PC1 values) drives increases in bacterial diversity. There was no corresponding relationship between diet and richness for either bacteria or fungi in birds. Shaded areas represent 95% credible intervals. (Online version in colour.)

between bacterial richness and the primary axis of a PCA of dietary variation (figure 1b). This indicates that bacterial alpha-diversity increases with progressively more vegetation, and less invertebrate prey, in the diet. We also found support for an effect of PC2 on both fungal and bacterial diversity, suggesting an increase in diversity in tandem with the proportion of dietary seeds and fruits, though the relationship was weaker (electronic supplementary material, figure S4). However, we found no relationship between diet and microbial diversity in birds across both fungi and bacteria (Figure 1b, credible intervals for slopes of all terms involving PC1 and PC2 all include zero).

### (ii) Beta-diversity

Patterns of variation in microbial community composition broadly followed those for alpha-diversity above. While for mammals, there was a significant correlation between host-associated bacterial community composition and diet ($r = 0.33$, $p = 0.002$) and a near-significant relationship between fungal community composition and diet ($r = 0.14$, $p = 0.067$), for birds, there was no significant relationship between dietary data and bacterial community composition ($r = 0.09$, $p = 0.211$) or fungal community composition ($r = 0.03$, $p = 0.386$). Further, taxonomic differences in microbiome composition based on differences in crude dietary patterns were not clear for either bacteria or fungi when the microbiome composition was visualized at the family level (electronic supplementary material, figures S5 and S6). That said, Alpha-proteobacteria and Eurotiomycete fungi were notably absent from species that primarily ate vegetation (i.e. grasses, etc), and Neocallimastigomycete fungi were the predominant fungal class associated with two out of four of these host species (electronic supplementary material, figure S5 and S6).

### (c) Strong evidence of correlated phylosymbiosis in both microbial groups

Our data revealed consistent variation in fungal and bacterial community structure across the host phylogeny (figure 2a). PERMANOVA analyses on centred-log ratio (CLR) transformed ASV abundances revealed significant phylogenetic effects of host class, order and species, as well as effects of sample storage and library preparation protocol for both microbial groups (table 1; electronic supplementary material, figures S7 and S8). For both bacteria and fungi, host species identity explained more variation than host class or order, and this pattern remained when re-running the models without sample preparation protocol effects, though this inflated the estimate of $R^2$ for all taxonomic groupings (table S2).

Consistent with our predictions, the similarity between the microbial communities of a given pair of host species was proportional to the phylogenetic distance between them (e.g. ASV level: fungal cor. = 0.26; $p = 0.001$; bacterial cor. = 0.37; $p = 0.001$; figure 2b). Correlations for both bacterial and fungal communities became stronger when aggregating microbial taxonomy to broader taxonomic levels (figure 2b). Notably, the bacterial correlation was stronger than the fungal equivalent at most taxonomic levels (figure 2b), indicating stronger patterns of phylosymbiosis for bacteria.

We also detected a strong, significant correlation between fungal and bacterial community structure of individual samples at the level of ASVs using Procrustes rotation (cor. = 0.29, $p < 0.001$; figure 2c). Collapsing ASV taxonomy to genus, family and order resulted in even stronger correlations (cor. = 0.44, 0.48 & 0.43, respectively; all $p < 0.001$; figure 2c). These data indicate a coupling between the structures of fungal and bacterial communities, whereby shifts in the structure of one community across the phylogeny also reflect consistent shifts in the other microbial group.

### (d) Patterns of co-occurrence of bacteria and fungi vary across host taxonomy

Analysis of correlations among fungal and bacterial abundances revealed differences in network structure at both the host class and host species level (electronic supplementary material, figures S9–S11). In particular, fungi of the phylum Ascomycota appeared frequently in the microbial networks of birds, mammals and amphibians (electronic supplementary material, figure S11). Models of species-level network data (electronic supplementary material, figures S9 and S10) revealed the frequency of positive co-occurrence between pairs of microbes also varied by class; Mammalia exhibited the highest proportion of positive edges (figure 3a), being significantly greater than those of birds (mean diff. 0.042 (0.017–0.067)) and amphibians (mean diff. 0.050 (0.002–0.112)). Notably, insects had a markedly lower proportion of positive edges compared to all other taxa (figure 3a). Class explained 93.2% (92.9–93.4%) of variation in edge sign. There was also systematic variation in network structure among taxonomic groups. Using the class-level network data (electronic supplementary material, figure S11), we estimated that Mammalia exhibited the fewest components, fewest communities and lowest modularity (electronic supplementary material, table S3 and figure 3b), indicating lower overall network subdivision relative to other animal classes, though this suite of traits is strongly correlated (electronic supplementary material, figure S12). Mean betweenness of fungal nodes also varied by host class; randomizations revealed that mean fungal betweenness was significantly lower than expected by chance in Aves ($p = 0.044$; figure 3c) but not Mammalia ($p = 0.600$; figure 3c). When investigating patterns of co-occurrence between fungal and bacterial taxa from the class-level networks, Aves displayed a significantly higher frequency of positive Actinobacteria–Ascomycota associations than expected by chance ($p = 0.002$; figure 3d). Though there was variation across host classes in which pair of phylum-level co-occurrences were most frequent, none occurred more significantly than expected by chance (all $p > 0.740$).

## 3. Discussion

Our study represents the most wide-ranging evaluation of animal mycobiome composition, and its covariation with the bacterial microbiome, undertaken to date. Our data provide novel evidence for mycobiome phylosymbiosis in wild animals, indicative of close evolutionary coupling between hosts and their resident fungal communities. Consistent with previous studies, we also find evidence of phylosymbiosis in the bacterial microbiome [29], but crucially, we demonstrate strong and consistent covariation between fungal and bacterial communities across host phylogeny, especially at higher microbial taxonomic levels. These patterns are supported by complementary network analysis illustrating frequent

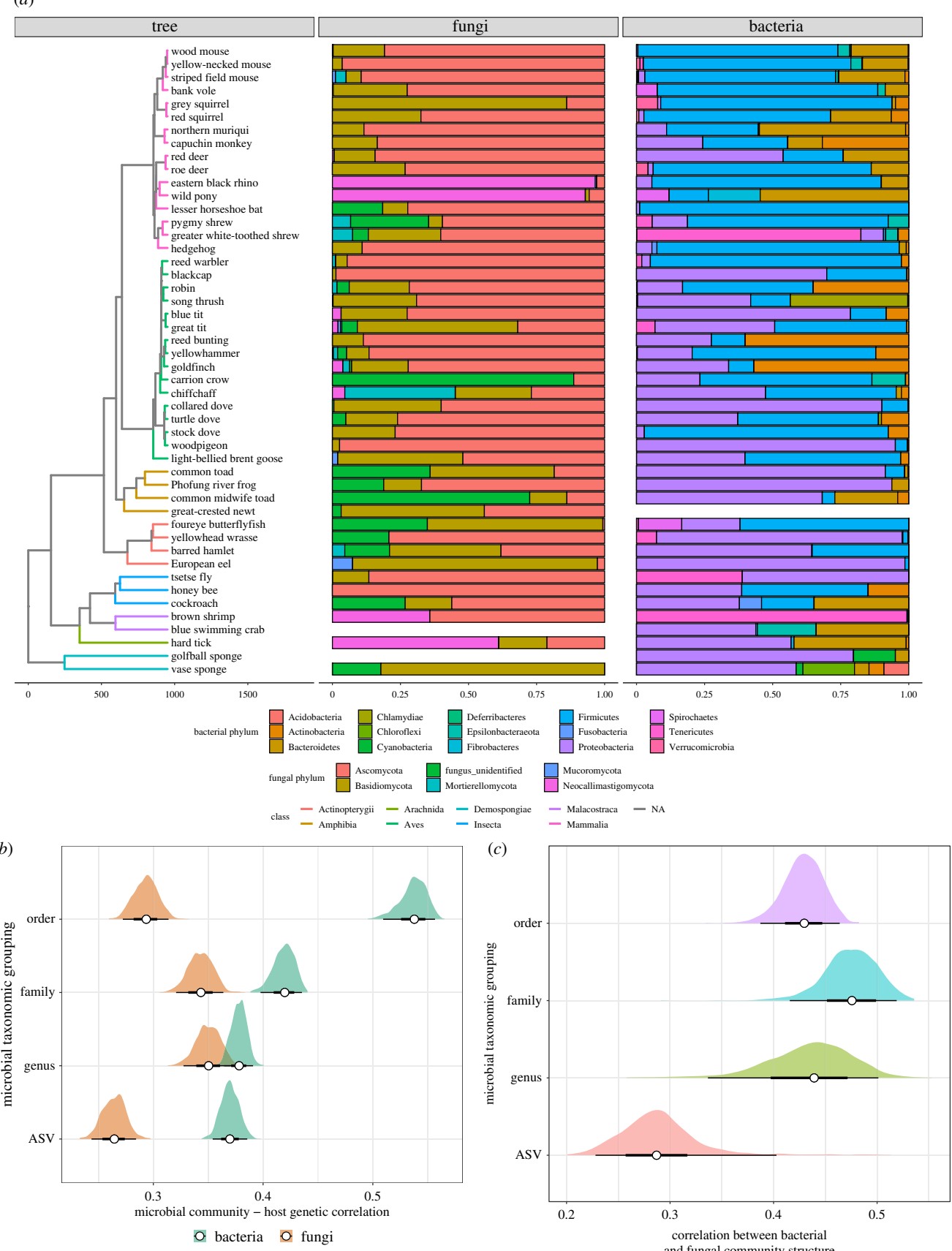

**Figure 2.** (a) Phylogenetic tree of host species, with branches coloured by class and node points coloured by order. Barplots show the proportional composition of fungal and bacterial phyla for each host species, aligned to tree tips. (b) Correlation between microbial and host genetic distances (generated from the phylogenetic tree in (a)) for both bacteria (green) and fungi (orange) across all host species. Microbial taxonomy was either raw ASVs or grouped into the family level. Aggregation to family resulted in higher correlations for both microbial groups, and the correlation was always stronger in bacteria. (c) Correlation between fungal and bacterial community structure derived from Procrustes rotation on PCA ordinations of each microbial group. Microbial communities were aggregated at various taxonomic groupings (order, family and genus), or as raw ASV taxonomy. For both (b) and (c), distributions of correlation values were generated using resampling of 90% of available samples for that microbial group to generate 95% intervals (shaded areas on graphs). Empty bars in (a) mean samples were not available for a particular species and so would not have been included in the calculations in (b) or (c). (Online version in colour.)

**Table 1.** PERMANOVA results for (*a*) fungi and (*b*) bacteria of factors explaining variation in microbial community structure. Terms were added in the order shown in the table to marginalize effects of sample storage and preparation protocols before calculating per cent variance explained for taxonomic groupings. Species ID was the dominant source of variation in the data for both taxonomic groups, but there were also strong effects of sample storage and wet laboratory protocol, particularly for bacteria.

| predictor | d.f. | $R^2$ | *p*-value |
|---|---|---|---|
| (*a*) FUNGI | | | |
| sample type | 7 | 0.05 | 0.001 |
| tissue storage | 5 | 0.04 | 0.001 |
| extraction kit | 7 | 0.07 | 0.001 |
| class | 2 | 0.02 | 0.001 |
| order | 6 | 0.05 | 0.001 |
| species | 18 | 0.09 | 0.001 |
| residuals | 303 | 0.68 | |
| (*b*) BACTERIA | | | |
| sample type | 6 | 0.06 | 0.001 |
| tissue storage | 6 | 0.16 | 0.001 |
| extraction kit | 7 | 0.12 | 0.001 |
| class | 2 | 0.02 | 0.001 |
| order | 6 | 0.09 | 0.001 |
| species | 18 | 0.12 | 0.001 |
| residuals | 273 | 0.42 | |

correlative links between fungal and bacterial taxa, whereby certain pairs of microbes from different kingdoms are much more likely to co-occur in the microbiome than expected by chance. Taken together, these data provide novel evidence of host recruitment for specific fungal and bacterial communities, which in turn may reflect host selection for interactions between bacteria and fungi critical for host physiology and health.

We found marked variation among host species in microbial community richness and composition for both bacteria and fungi. Though our data suggest many species support a diverse assemblage of host-associated fungi, we show that bacterial diversity tends to be higher on average relative to fungal diversity, and that there is no signal of positive covariance between fungal and bacterial richness within species, suggesting more ASV-rich bacterial microbiomes are not consistently associated with more ASV-rich mycobiomes. These patterns could arise because of competition for niche space within the gut, where high bacterial diversity may reflect stronger competition that prevents the proliferation of fungal diversity. Understanding patterns of niche competition within and among microbial groups requires that we are able to define those niches by measuring microbial gene function and quantifying the degree of overlap or redundancy in functional genomic profiles across bacteria and fungi.

We detected strong phylosymbiosis for both fungi and bacteria across a broad host phylogeny encompassing both vertebrate and invertebrate classes. This pattern was significantly stronger in bacteria than for fungi. In both microbial kingdoms, the signal of phylosymbiosis strengthened when aggregating microbial assignments to broader taxonomic levels, a phenomenon that has previously been shown for bacterial communities [30]. That this pattern also occurs in fungi suggests either that host recruitment is weaker at finer-scale taxonomies, or our ability to detect that signal is weaker at the relatively noisy taxonomic scale of ASVs. Stronger signals of phylosymbiosis at family and order-level taxonomies may reflect the deep evolutionary relationships between hosts and their bacterial and fungal communities, as well as the propensity for microbial communities to allow closely related microbes to establish while repelling less related organisms [31]. That is, higher order microbial taxonomy may better approximate functional guilds within the microbiome, such as the ability to degrade cellulose [25,30], which are otherwise obscured by taxonomic patterns of ASVs. Resolving this requires the integration of functional genomic data from the fungal and bacterial microbiota into the phylogeny.

Network analyses of microbial co-occurrence supported our findings of microbe-specific patterns of phylosymbiosis by revealing strong covariation between fungal and bacterial community composition across the host phylogeny. These patterns are consistent with host recruitment for particular suites of fungal and bacterial taxa, which may represent bacteria–fungi metabolic interactions beneficial to the host. Bacterial–fungal interactions have previously been demonstrated for a handful of animal species [8,9,17,32,33], but here we show these could be widespread across multiple animal classes. Both bacteria and fungi have considerable enzymatic properties that facilitate the liberation of nutrients for use by other microbes, thus facilitating cross-kingdom colonization [34–36] and promoting metabolic inter-dependencies [37–39]. The frequency and predicted direction of co-occurrence relationships varied considerably among host classes, with the mammalian network exhibiting (i) a lower modularity, indicating weaker clustering into fewer discrete units (both distinct components and interlinked communities) and (ii) a higher frequency of positive correlations between microbes (at the host species level) compared to most other classes, in particular birds and insects. Higher modularity in Aves compared to Mammalia could represent stronger associations within distinct suites of microbes, compared to a more homogeneous network in the latter. Similarly, we found that fungal betweenness was significantly lower than expected by chance, suggesting that fungal nodes are relatively more peripheral within these clusters in Aves compares to Mammalia. This could mean that fungi are less important as 'hubs' of putative metabolic networks in birds and therefore less likely to be form links to other clusters. Comparisons of networks are challenging when they differ in size (i.e. number of nodes) and structure, and differences between classes in traits like modularity and betweenness will also be affected by species replication within each class and factors like the ecological breadth of hosts. Even if the species-level sample size is identical across classes, lower modularity could be expected to arise if hosts in one class were more similar in terms of ecological niche or traits like diet. However, proportional traits like interaction structure (proportion of positive interactions) are unlikely to be driven solely by sample size, suggesting marked biological variation in strength of fungi–bacteria relationships across the host phylogeny. This is particularly evident in Aves, where we observed a high frequency of positive associations between Ascomycota and Actinobacteria.

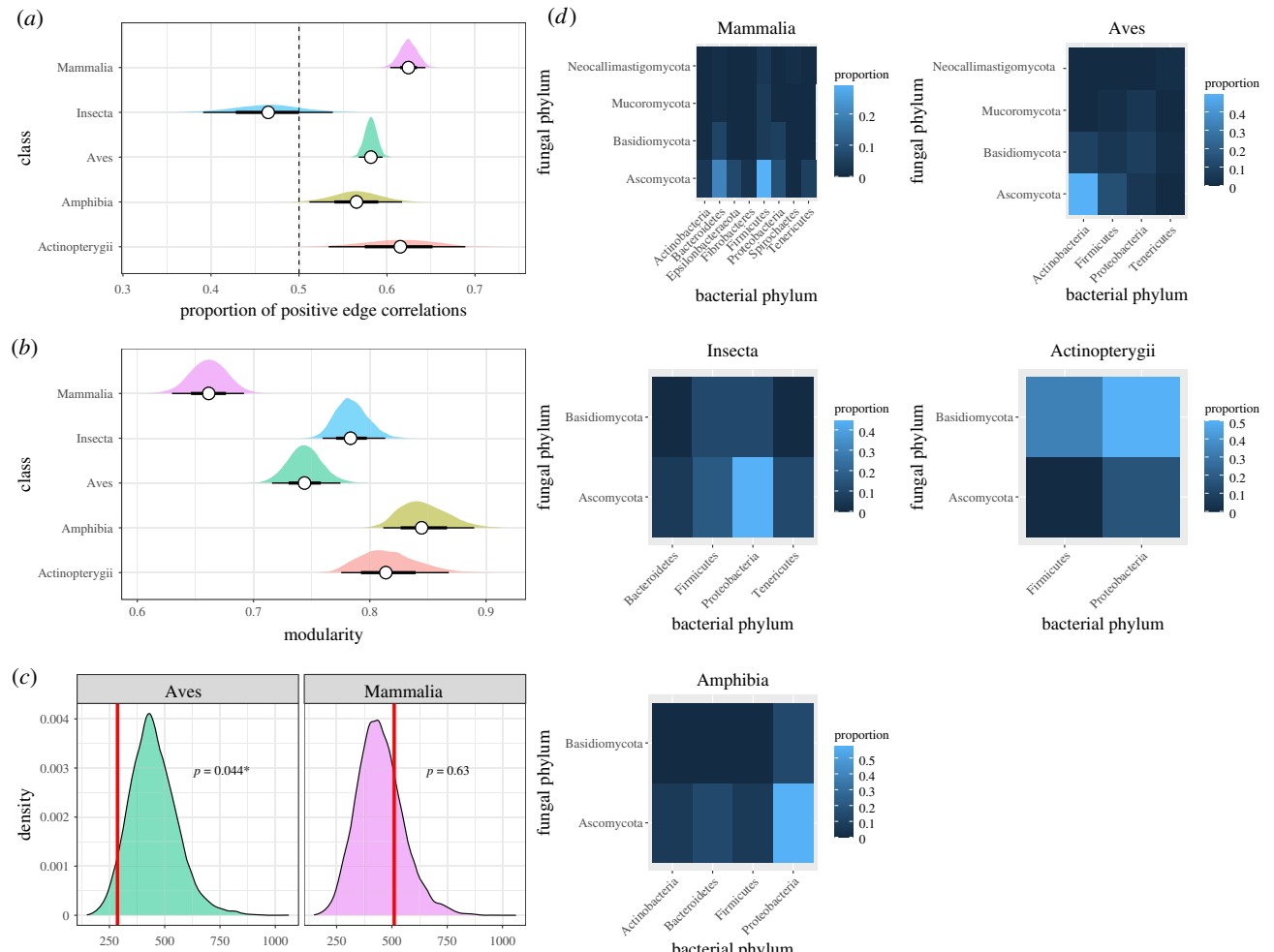

**Figure 3.** (a) Analysis of network structural traits from species-specific networks comprising 39 species from five classes. There were significant differences in the proportion of positive edges (correlations between paired microbial abundance values) among classes. Vertical dashed line indicates equal proportion of positive and negative edges. (b) Modularity scores from class-specific networks. For both (a) and (b), white points indicate means, black bars are 66% and 95% intervals, and shaded areas are the full sampling distribution. (c) Permutational testing revealed that mean fungal betweenness was significantly lower than expected by chance in Aves, but not Mammalia, indicating heterogeneity in network structure. (d) Heatmaps of frequencies of co-occurrence (proportion of positive edges) between fungal and bacterial taxa at the phylum level split by host class. Permutation testing revealed that coupling of Ascomycota and Actinobacteria in Aves occurred significantly more often than expected by chance. Note class-specific heatmap scales. (Online version in colour.)

Actinobacteria are crucial for the maintenance of gut homeostasis [40], whereas the increased prevalence of members of Ascoymycota have been associated with both healthy and impaired gut function in humans [41]. That we observe such a tight coupling between these two cross-kingdom taxonomic groups suggests they may be involved in cross-feeding metabolic networks within the avian gut, or alternatively may be in competition for similar niche space. It is important to note that positive correlations between microbial abundances are not themselves evidence of interactions between those species. We suggest they represent novel hypotheses that could be tested in controlled systems, where microbiome composition and therefore the interactions among microbes can be manipulated to test the influence of such interactions on host physiology.

The drivers of phylosymbiosis remain unclear, even for bacterial communities; is a phylogenetic signal indicative of host–microbiome coevolution, or simply a product of 'ecological filtering' of the microbiome in the host organism either via extrinsic (e.g. diet, habitat) or intrinsic sources (e.g. gut pH, immune system function) [26,29,42]? Our results indicate host diet may play a role in determining bacterial

and fungal composition in mammals, but not in birds. These results are broadly consistent with previous work, where the influence of diet on bacterial microbiome was most evident in mammals [25], or only present at the species level in New Guinea passerines [43]. For example, the composition and diversity of both fungal and bacterial communities of faecal samples differed between phytophagous and insectivorous bats [16], and the fungal community composition of mice guts were affected by fat content of their diet [17]. However, recent work has revealed diet–microbiome correlations in non-passerine birds [44], suggesting the relationship may not be uniform across finer taxonomic scales within host classes. It is also worth noting that the signals produced from faecal and true gut samples may differ; evidence suggests faecal samples may indicate diet is the predominant driver of 'gut' microbiome composition when gastrointestinal samples indicate host species is the predominant determinant [45]. Moreover, faecal samples may only represent a small proportion of the gastrointestinal microbiome [45–47].

Our data also show that sample type has a significant effect on both fungal and bacterial community composition. In addition, DNA extraction kit and storage method also

affected microbiome composition, as seen in previous work [48–51]. As our study made use of existing samples, we had limited ability to influence the sample type, storage and DNA extraction method, and in some cases, these varied considerably even within a class (e.g. across the four amphibian species used, we had four different tissue types). We also had considerable variation in approaches to sample storage across the study species, including freezing, storage in ethanol and storage in RNA later, and a variety of different extraction kits with different approaches to lysis (both mechanical and chemical), which will ultimately affect which microbial taxa are detected. Despite these limitations, we still find compelling evidence for phylosymbiosis in fungal communities, and strong associations between bacterial and fungal taxa across a broad range of primarily wild host species. However, a more thorough analysis of true gut communities is required to fully characterize mycobiome phylosymbiosis and dietary signals across wild animals, and to identify other ecological and host-associated factors that influence mycobiome composition and function. We hypothesize that evolutionary processes play a large role in shaping host-associated microbiomes, with selection for microbiome function rather than taxonomic groupings *per se*.

Within animals, the roles of host-associated fungal communities are not well understood, yet our data highlight that fungi are important components of microbiome structure that are often overlooked. Our knowledge of the range of functions provided by the host mycobiome, and how these alter or complement those provided by the bacterial microbiome remain limited. We hypothesize that host-associated fungi and bacteria produce mutually beneficial metabolites that facilitate the colonization, reproduction and function of cross-kingdom metabolic networks [28]. Though we provide evidence for consistent variation among host class in fungal community structure, and the role of fungi within microbial co-occurrence networks, for many researchers, the questions of key interest will focus on what governs variation at the level of the individual. Clear gaps in our knowledge remain regarding the relative contributions of host genomic [52–54] and environmental variation to host mycobiome structure, function and stability. There is an urgent need to incorporate quantitative estimates of microbial function into microbiome studies, which are crucial for understanding the forces of selection shaping host–microbe interactions at both the individual and species level.

# 4. Material and methods

## (a) Sample collection

DNA was extracted from tissue or faecal samples of 49 host species using a variety of DNA extraction methods (electronic supplementary material, table S1) and normalized to approximately 5–10 ng μl$^{-1}$. Samples were largely collated from previous studies and/or those available from numerous researchers and as such, DNA extraction and storage techniques were not standardized across species. We sequenced a median of 10 samples per species (range of 5 to 12; electronic supplementary material, table S1).

## (b) ITS1F-2 and 16S rRNA amplicon sequencing

Full details are provided in the electronic supplementary material. Briefly, we amplified the ITS1F-2 rRNA gene to identify fungal communities using single index reverse primers and a modified protocol of Smith & Peay [55] and Nguyen *et al.* [56], as detailed in Griffiths *et al.* [13]. To identify bacterial communities, we amplified DNA for the 16S rRNA V4 region using dual indexed forward and reverse primers according to Kozich *et al.* [57] and Griffiths *et al.* [53]. The two libraries were sequenced separately using paired-end reads (2 × 250 bp) with v2 chemistry on an Illumina MiSeq.

We conducted amplicon sequence data processing in DADA2 v. 1.5 [58] in RStudio v. 1.2.1335 for R [59,60] for both ITS rRNA and 16S rRNA amplicon data. After data processing, we obtained a median of 1425 reads per sample (range of 153 to 424 527) from the ITS data and a median of 3273 reads (range of 153 to 425 179) for the 16S rRNA data.

To compare alpha-diversity between species and microbial kingdoms, we rarefied libraries to 500 reads per sample, yielding 292 samples from 46 species and 307 samples from 47 species for fungal and bacterial kingdoms, respectively. Alpha-diversity measures remained relatively stable within a host species whether data were rarefied to 500, 1000, or 2500 reads (figure 1, electronic supplementary material, figure S1; see electronic supplementary material for more details).

## (c) Host phylogeny

We built a dated phylogeny of host species using TimeTree [61] using 42 species, of which 36 were directly represented in the TimeTree database. A further six species had no direct match in TimeTree and so we used a congener as a substitute (*Amietia*, *Glossina*, *Portunus*, *Ircinia*, *Amblyomma* and *Cinachyrella*). We calculated patristic distance among species based on shared branch length in the phylogeny using the 'cophenetic' function in the *ape* package [62]. We visualized and annotated the phylogeny using the R package *ggtree* [63]. To create a phylogeny for all samples, we grafted sample-level tips onto the species phylogeny with negligible branch lengths following Youngblut *et al.* [25].

## (d) Fungal and bacterial community analysis

We used *brms* [64,65] to fit (generalized) linear mixed-effects models [(G)LMMs] to test for differences in alpha-diversity and calculated $r^2$ of models using the 'bayes_R2' function. We assessed the importance of terms based on whether 95% credible intervals of the parameter estimates crossed zero. We used *ggplot* [66], *cowplot* [67] and *tidybayes* [68] for raw data and plotting of posterior model estimates. To support these analyses, we used *phylobase* [69] and *phylosignal* [70] to estimate the phylogenetic signal in patterns of alpha-diversity for both bacteria and fungi, using both Inverse Simpson Index and number of observed ASVs as outcome variables. We calculated Abouheif's C$_{mean}$ for each diversity–microbe combination and corrected *p*-values for multiple testing using Benjamini–Hochberg correction.

To identify taxonomic differences in microbiome and mycobiome composition between host species, we used CLR transformation in the *microbiome* [71] package to normalize microbial abundance data, which obviates the need to lose data through rarefying [72]. To quantify differences in beta-diversity among kingdoms and species while simultaneously accounting for sample storage and library preparation differences among samples, we conducted a PERMANOVA analysis on among-sample Euclidean distances of CLR-transformed abundances using the *adonis* function in *vegan* [73] with 999 permutations. For both kingdoms, we specified effects in the following order: sample type, tissue storage, extraction kit, class, order, species. This marginalizes the effects of sample metadata variables first, before partitioning the remaining variance into that accounted for by host phylogeny. The results were similar when amplicon data were converted to the relative abundance or rarefied to 500 reads (electronic supplementary material, table S4).

To test the hypothesis that inter-individual differences in microbial community composition were preserved between microbial kingdoms, we performed Procrustes rotation of the two PCA ordinations for bacterial and fungal abundance matrices, respectively ($n = 277$ paired samples from 46 species). We also repeated this analysis with ASVs agglomerated into progressively higher taxonomic rankings from genus to order [30]. To provide a formal test of differences in strength of covariation at different taxonomic levels, we conducted a bootstrap resampling analysis where for each kingdom at each iteration, we randomly sampled 90% of the data and recalculated the correlation metric. We repeated this process 999 times to build a distribution of correlation values at each taxonomic grouping. To examine the hypothesis that inter-individual distance in microbial community composition varies in concert with interspecific phylogenetic distance, we performed a Procrustes rotation on the paired matrix of microbial distance (Euclidean distance of CLR-transformed abundances) and patristic distance from the phylogenetic tree.

To identify potential co-occurrence relationships between fungal and bacterial communities, we conducted two analyses; (i) we used the *SpiecEasi* [74] to identify correlations between unrarefied, CLR-transformed ASVs abundances at the host class level (with insects grouped), and (ii) we used co-occurrence analysis at the species level, by rarefying the bacterial and fungal datasets to 500 reads each, and agglomerated taxonomy family level, resulting in 117 bacterial groups and 110 fungal groups. These analyses are complementary as *cooccur* is more sensitive and *SpiecEasi* more specific in terms of the ability to detect statistical associations between pairs of microbes. Both methods identified significant variation in network composition between host class (i.e. when species-level data were scaled up to class level, figure 3a). We then merged the *phyloseq* objects for bacterial and fungal communities for each sample, with sufficient data retained to conduct the co-occurrence analysis for 40 host species. Using these cross-kingdom data, we calculated the co-occurrence between each pair of microbial genera by constructing a Spearman's correlation coefficient matrix in the *bioDist* package [75,76]. We visualized those with $\rho$ greater than 0.50 (strong positive interactions) and $\rho < -0.50$ (strong negative interactions) for each host species separately using network plots produced in *igraph* [77].

To determine the effect of diet on bacterial and fungal community composition, we used only samples from the bird and mammal species and agglomerated the data for each host species using the merge_samples function in *phyloseq* [78]. We obtained dietary data for each host species from the EltonTraits database, which provides standardized and semi-quantitative diet data for host species based on descriptions from global handbooks and monographs [79]. Further details on network and dietary analyses are in electronic supplementary material.

**Ethics.** This research made use of samples or DNA previously collected for other studies, in accordance with local legislation and UK ethical standards. Where necessary, the research was additionally approved by the ethics committee at the University of Salford. Further details of specific permits and licenses are available in the electronic supplementary material.

**Data accessibility.** Sequence data are deposited in the NCBI SRA database under BioProject nos. PRJNA593927 and PRJNA593220. A fully reproducible analysis workflow can be accessed at https://github.com/xavharrison/Mycobiome2020.

The data are provided in the electronic supplementary material [80].

**Authors' contributions.** X.A.H.: formal analysis, writing-original draft, writing-review & editing; A.M.: resources, writing-review and editing; J.D.: resources, writing-review and editing; S.M.G.: resources, writing-review and editing; C.B.: resources, writing-review & editing; R.B.: resources, writing-review and editing; J.P.B.: resources, writing-review and editing; K.B.: resources, writing-review and editing; C.B.: resources, writing-review and editing; D.R.B.: resources, writing-review and editing; S.S.B.: resources, writing-review and editing; R.C.: resources, writing-review and editing; J.C.: resources, writing-review and editing; F.C.: resources, writing-review and editing; I.C.: resources, writing-review and editing; K.L.E.: resources, writing-review and editing; N.F.: resources, writing-review and editing; I.G.: resources, writing-review and editing; A.H.: resources, writing-review and editing; J.H.: resources, writing-review and editing; J.J.: resources, writing-review and editing; R.J.: resources, writing-review and editing; M.K.: resources, writing-review and editing; T.K.: resources, writing-review and editing; J.M.D.L.: resources, writing-review and editing; J.L.L.: resources, writing-review and editing; A.M.: resources, writing-review and editing; J.M.: resources, writing-review and editing; B.M.: resources, writing-review and editing; D.O.: resources, writing-review and editing; P.P.: resources, writing-review and editing; R.F.P.: resources, writing-review and editing; J.K.R.: resources, writing-review and editing; S.S.: resources, writing-review and editing; M.S.: formal analysis, resources, writing-review and editing; J.E.S.: resources, writing-review and editing; W.O.C.S.: resources, writing-review and editing; M.P.: resources, writing-review and editing; S.L.W.: resources, writing-review and editing; M.D.W.: resources, writing-review and editing; E.M.: writing-review and editing; R.A.: conceptualization, data curation, formal analysis, project administration, resources, writing-original draft, writing-review and editing.

All authors gave final approval for publication and agreed to be held accountable for the work performed therein.

**Competing interests.** The authors have no competing interests to declare.

**Funding.** There was no specific funding attached to this project, but previous sample collection was supported by the following: Devenish Nutrition funded the red deer research in Ireland. Fieldwork to collect avian faecal samples was funded by The Royal Society, the RSPB and Natural England. Small mammal sampling was supported by NERC, Radioactive Waste Management Ltd and the Environment Agency and the Genetics Society. Northern muriqui research was funded by CAPES, Brazil.

**Acknowledgements.** We would like to thank Miran Aprahamian, Chris Williams (Environment Agency) Patrick Abila (National Livestock Resources Research Institute) and Dr Patrick Vudriko (Makerere University) for providing samples, as well as BEI Resources, the US Forest Service and the University of Wisconsin-Madison for providing mock communities.

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
