## [Peer Review File · Proceedings of the Royal Society B: Biological Sciences]

Review History

RSPB-2021-0552.R0 (Original submission)

Review form: Reviewer 1

Recommendation

Major revision is needed (please make suggestions in comments)

Scientific importance: Is the manuscript an original and important contribution to its field?

Acceptable

General interest: Is the paper of sufficient general interest?

Acceptable

Quality of the paper: Is the overall quality of the paper suitable?

Acceptable

Is the length of the paper justified?

Yes

Should the paper be seen by a specialist statistical reviewer?

Yes

Do you have any concerns about statistical analyses in this paper? If so, please specify them explicitly in your report.

No

It is a condition of publication that authors make their supporting data, code and materials available - either as supplementary material or hosted in an external repository. Please rate, if applicable, the supporting data on the following criteria.

Is it accessible?

Yes

Is it clear?

Yes

Is it adequate?

Yes

Do you have any ethical concerns with this paper?

No

Comments to the Author

In their manuscript Xavier and colleagues use amplicon sequencing to quantify the bacterial and fungal communities within a range of hosts. They use various phylogenetic analyses to identify the role of host phylogeny in shaping microbial community composition, assess the impact of diet on microbial richness, and construct bacteria-fungi correlation / co-occurrence networks within different hosts. They argue their paper shows the clear need to look beyond bacteria when studying host-associated microbiota.

Overall this is an interesting paper - fungi are consistently ignored from most microbiome analyses so I welcome any work addressing this gap. I'm not an expert in phyllosymbioses, so can't comment on the validity of these analyses, but the results (that fungal and bacterial communities covary across hosts, but to different extents) are interesting and good to see laid out, even if they're not necessarily surprising.

My major concern lies with the network analysis and its interpretation. First, crucially, the network analyses used uncover correlations / co-occurrences between different taxa. What they don't tell us are whether / how these taxa are actually interacting. This doesn't mean these analyses aren't meaningful or interesting, but this distinction needs to be made clearer both in the initial outline of the analyses and in their interpretation. Currently at a number of points in the manuscript (including the abstract) the authors imply they are determining the strength / sign / frequency of microbe-microbe interactions.

More broadly, I think a lot more could be made of the network analysis, both in what's analysed and in its interpretation. For example:

- What's the biological meaning of these betweenness or modularity measures? Are you implying some networks are more "hub-y" than others, or that there tend to be distinct community states? I would appreciate some interpretation here as otherwise they're just generic network properties
- Can you make more of this novel fungal data? Are fungal communities structured any differently to bacterial ones (for example, are there any differences in proportion of positive/negative correlations than we might expect by chance? You touch on this very briefly in the caption of Fig 3B but I think more could be made of this). Similarly, you note that there are correlations between specific fungi and bacteria, but can you say something more about this? For example, do specific fungi/bacteria often co-occur across taxa?
- Re the proportion of positive:negative correlations, is this just reflecting differences in diversity between the different taxa? It will higher diversity definitionally lead to increases in positive correlations?
- From a technical perspective, you seem to be using two approaches for determining cooccurrences between taxa (line 401) but it's not clear to me why or what differences these methods yielded – please can you clarify?

Smaller comments

Fig 1A: I appreciate having raw datapoints on these plots, but currently they obscure the box and whiskers making comparisons hard. Can you adjust this to increase readability, eg by making dots smaller / transparent / something.

Fig 1A How do the amphibia etc plots look if they're on the same y-scale? Ideally I'd like to be able to compare directly between plots, though I appreciate this might not be practical.

Can you give more of an explanation in the text of the diet metric used? Might also be worth adding a "increasing plant material" label to the axis of fig 1B to help interpretability. Related, is there any relationship between richness and diversity for the other principal components? I'm not sure Fig 3A really adds much, perhaps this panel could be used for some more informative network metrics?

Review form: Reviewer 2 (Andrew Brooks)

Recommendation

Accept with minor revision (please list in comments)

Scientific importance: Is the manuscript an original and important contribution to its field?

Excellent

General interest: Is the paper of sufficient general interest?

Excellent

Quality of the paper: Is the overall quality of the paper suitable?

Acceptable

Is the length of the paper justified?

Yes

Should the paper be seen by a specialist statistical reviewer?

No

Do you have any concerns about statistical analyses in this paper? If so, please specify them explicitly in your report.

No

It is a condition of publication that authors make their supporting data, code and materials available - either as supplementary material or hosted in an external repository. Please rate, if applicable, the supporting data on the following criteria.

Is it accessible?

Yes

Is it clear?

Yes

Is it adequate?

No

Do you have any ethical concerns with this paper?

No

Comments to the Author

Primary Concern:

Given the challenges of acquiring samples from mostly wild species it is understandable that there would be some variation in collection and storage methods, and I think the use of mostly wild organisms is a major strength of this paper. However, variation in specimen type and storage are pronounced across species, and DNA extraction methods varied dramatically across species as well. Supplementary PDF File 1 – Line 699 Table S1: DNA was obtained from a variety of different sources, which the authors have compiled in supplementary table 1. It is an admirable job noting all of the details and I commend the authors on the thorough work. As the main motivation of this paper is interspecific comparisons, this is my primary concern in accepting the validity of their conclusions. The sample origin, storage, and processing vary in at least one factor between most species being compared, raising the question of whether interspecific variation is biological or methodological. The authors try to address this in Table 2 with a sequential multivariate PERMANOVA analysis including sample type, tissue storage, and extraction kit as covariates prior to host taxonomic predictors. The authors also do mention these limitations on Lines 303 – 312 at the end of the discussion which I commend.

At a minimum I think the authors should clearly state these limitations strongly up front in the paper (abstract or introduction), primarily because the issues are more pronounced compared to other studies. This could be balanced by a clearer argument of the strengths of obtaining samples from wild organisms in natural habitats, which justifies some of the variation in approach. A more thorough discussion of likely effects of differences in specimen type (whole organism, stool, extracted gut, toe clipping...), storage medium, between kit variation, and lysis approach would help with details from the literature. The ideal scenario would be benchmarking differences between these approaches, but seems beyond the scope of this study.

Some key examples :

A. Samples within the same host class were of very different tissues. For instance some insects (e.g. cockroaches, honeybees) had guts extracted while others used the whole organism for DNA extraction (e.g. tsetse fly, hard tick). A more concerning example is amphibia where among four species sampled, one species extraction used a skin swab, one used dissected tadpole mouthparts, one used the whole organism, and one used a toe clipping. Birds and mammals tend to be more consistent using either dissected guts or fecal samples, and therefore relationships are likely more consistent with biology.

B. Tissue storage medium is known to bias microbiome profiles considerably as the authors

note at line 309, and there was significant variation including: raw frozen samples, samples stored in varying concentrations of ethanol, and samples stored in RNALater.

C. As many species samples are extracted at different sites and with different kits, there is also likely critical variation in the lysing methods of samples. Whether bead beating (either in a proper bead beating machine or desktop vortexer), through chemical lysis, or mechanical lysis can play an enormous role on the detection of certain microbial partners, and while not noted these likely varied between different sites performing extractions.

I think this manuscript has significant value to the field, and this primary concern is not easily remedied without extensive field and lab work for identical sample processing. Therefore I think the best course would be to make this limitation very clear up front and discuss how individual factors may affect the analyses. Ultimately I think the strengths in bacteria to fungal comparison and use of wild animals outweigh the potential biases in sample handling, as long as the reader can clearly make this judgement for themselves.

Data Accessibility:

The SRA sequence archive is accessible and all samples appear clearly labeled by mycobiome and microbiome, and the github repository contains a well documented r-markdown script of downstream analyses. I was unable to locate the key metadata and dietary files, and could not locate scripts relating to the processing / quality control of raw reads, establishment into ASVs, and taxonomic assignment with DADA2. The r-markdown script appears to start with established community profiles, but scripts for processing raw sequences into profiles could not be located.

Additional points:

Figure 1A: Using inverse Simpson diversity is an appropriate alpha diversity measure, however it is likely not the best choice for the comparisons they are making and the display of this data. The issue is that the tables are rarefied to 500 reads per sample and with Simpson diversity this is likely driving the outlying values. A few individual samples have high diversity (index >35) while most are <5, leaving most species clustered at the bottom which makes between species variation difficult to examine. A better index to draw out the spread in diversity values between species would be Shannon's diversity index. Shannon also might be more appropriate for the low coverage and very different compositions of the microbiomes across species as it has a logarithmic transformation in the calculation which weights rarer organisms more heavily. Making distinguishing between species different more difficult is the overlay of individual samples with opaque white circles bordered in green or orange, as they frequently obscure the quartile and median black bars of the boxplots. Finally, the authors should clarify in the figure legend (like figure S1A: Median (\pm 25th and 75th percentiles)) what parts of the boxplot detail as some programs use different values (quartiles, mean versus median of central bar, that the circles are individual sample values).

Supplementary Methods: PCR amplification was done with 30 cycles which is somewhat high given the 10ng/uL input DNA concentration, which could increase PCR artifacts and off-target amplification. Maybe mention why 30 cycles was chosen.

Line 193-207, 225, 270-290: This network analysis is creative and informative. The repeated comparison that mammals have highly connected networks and insects and birds much less so, however is the analytical result I would trust the least in this paper given the variation in sample collection and processing noted in supplementary table 1 (insects and birds have much higher variability than mammals). While this would be the biological pattern I would expect, I think the authors need to clearly note in discussion Line 280 that these patterns could also be influenced by sampling biases between host classes.

Line 256-269: This is an excellent discussion of phyllosymbiosis variation at different microbial taxonomic cutoffs.

Line 350: Samples were rarefied to 500 reads per sample for alpha diversity analyses, which is low coverage. The authors did verify that alpha diversity remained relatively stable when sampling at 1,000 or 2,500 reads (Figure S1). It appears that samples were rarefied a single time at each level, best practice would be to repeatedly rarefy at each sequence cutoff (e.g. rarefied 100 times and the mean alpha diversity values used across the rarefactions). I don't think they would see much variation in results however.

Line 378: The use of CLR and euclidian distance for beta diversity is an interesting choice, but likely meant to avoid loss of samples in rarefying cutoffs. Line 387: I think showing that results are similar with rarefaction should be presented in the supplement if it is being mentioned in the manuscript. The use of "(data not presented)" does not seem adequate here. If analyses are done with rarefaction on samples not CLR transformed I am wondering what distance metric was used (or were they CLR transformed after rarefying and then euclidian used)? It would be ideal to see an alternate metric like Jaccard distance or Bray-Curtis validated on the non-CLR transformed rarefied tables for the same PERMANOVA analysis in Table 2.

Summary: I found the analyses to be thorough, well executed, and proper to address the hypothesis in question. The primary weakness of the paper seems to result from sampling that was beyond the authors control for the most part, and is documented in detail in table S1. I think it should be more clearly stated early on as a potential weakness (given it cannot be easily remedied) so the reader can judge for themselves. Use of wild animals compared to lab / zoo animals is an enormous strength in my opinion, and the comparison of fungal and bacterial communities across so many host species is completely novel. The writing is clear and focused, and the discussion was thought provoking and robust. Overall this was quite an enjoyable and informative paper to read.

Decision letter (RSPB-2021-0552.R0)

20-Apr-2021

Dear Dr Antwis:

Your manuscript has now been peer reviewed and the reviews have been assessed by an Associate Editor. The reviewers' comments (not including confidential comments to the Editor) and the comments from the Associate Editor are included at the end of this email for your reference. As you will see, the reviewers and the Editors have raised some concerns with your manuscript and we would like to invite you to revise your manuscript to address them.

Research ethics:

Use of animals and field studies:

It is a condition of publication that you make available the data and research materials supporting the results in the article. Please see our Data Sharing Policies (<https://royalsociety.org/journals/authors/author-guidelines/#data>). Datasets should be deposited in an appropriate publicly available repository and details of the associated accession number, link or DOI to the datasets must be included in the Data Accessibility section of the article (<https://royalsociety.org/journals/ethics-policies/data-sharing-mining/>). Reference(s) to datasets should also be included in the reference list of the article with DOIs (where available).

Please submit a copy of your revised paper within three weeks. If we do not hear from you within this time your manuscript will be rejected. If you are unable to meet this deadline please let us know as soon as possible, as we may be able to grant a short extension.

Best wishes,
Dr Maurine Neiman
mailto:proceedingsb@royalsociety.org

Associate Editor

Comments to Author:

We have obtained two detailed reviews from experts in the field. Both see merit in your study but raise a range of issues, both technical and conceptual, that require your response and some changes to the manuscript.

Reviewer(s)' Comments to Author:

Referee: 1

Comments to the Author(s)

In their manuscript Xavier and colleagues use amplicon sequencing to quantify the bacterial and fungal communities within a range of hosts. They use various phylogenetic analyses to identify the role of host phylogeny in shaping microbial community composition, assess the impact of diet on microbial richness, and construct bacteria-fungi correlation / co-occurrence networks within different hosts. They argue their paper shows the clear need to look beyond bacteria when studying host-associated microbiota.

Overall this is an interesting paper – fungi are consistently ignored from most microbiome analyses so I welcome any work addressing this gap. I'm not an expert in phyllosymbioses, so can't comment on the validity of these analyses, but the results (that fungal and bacterial communities covary across hosts, but to different extents) are interesting and good to see laid out, even if they're not necessarily surprising.

My major concern lies with the network analysis and its interpretation. First, crucially, the network analyses used uncover correlations / co-occurrences between different taxa. What they don't tell us are whether / how these taxa are actually interacting. This doesn't mean these analyses aren't meaningful or interesting, but this distinction needs to be made clearer both in the initial outline of the analyses and in their interpretation. Currently at a number of points in the manuscript (including the abstract) the authors imply they are determining the strength / sign / frequency of microbe-microbe interactions.

More broadly, I think a lot more could be made of the network analysis, both in what's analysed and in its interpretation. For example:

- What's the biological meaning of these betweenness or modularity measures? Are you implying some networks are more "hub-y" than others, or that there tend to be distinct community states? I would appreciate some interpretation here as otherwise they're just generic network properties
- Can you make more of this novel fungal data? Are fungal communities structured any differently to bacterial ones (for example, are there any differences in proportion of positive/negative correlations than we might expect by chance? You touch on this very briefly in

the caption of Fig 3B but I think more could be made of this). Similarly, you note that there are correlations between specific fungi and bacteria, but can you say something more about this? For example, do specific fungi/bacteria often co-occur across taxa?

- Re the proportion of positive:negative correlations, is this just reflecting differences in diversity between the different taxa? It will higher diversity definitionally lead to increases in positive correlations?

- From a technical perspective, you seem to be using two approaches for determining cooccurrences between taxa (line 401) but it's not clear to me why or what differences these methods yielded - please can you clarify?

Smaller comments

Fig 1A: I appreciate having raw datapoints on these plots, but currently they obscure the box and whiskers making comparisons hard. Can you adjust this to increase readability, eg by making dots smaller / transparent / something.

Fig 1A How do the amphibia etc plots look if they're on the same y-scale? Ideally I'd like to be able to compare directly between plots, though I appreciate this might not be practical.

Can you give more of an explanation in the text of the diet metric used? Might also be worth adding a "increasing plant material" label to the axis of fig 1B to help interpretability. Related, is there any relationship between richness and diversity for the other principal components?

I'm not sure Fig 3A really adds much, perhaps this panel could be used for some more informative network metrics?

Referee: 2

Comments to the Author(s)

Primary Concern:

Given the challenges of acquiring samples from mostly wild species it is understandable that there would be some variation in collection and storage methods, and I think the use of mostly wild organisms is a major strength of this paper. However, variation in specimen type and storage are pronounced across species, and DNA extraction methods varied dramatically across species as well. Supplementary PDF File 1 - Line 699 Table S1: DNA was obtained from a variety of different sources, which the authors have compiled in supplementary table 1. It is an admirable job noting all of the details and I commend the authors on the thorough work. As the main motivation of this paper is interspecific comparisons, this is my primary concern in accepting the validity of their conclusions. The sample origin, storage, and processing vary in at least one factor between most species being compared, raising the question of whether interspecific variation is biological or methodological. The authors try to address this in Table 2 with a sequential multivariate PERMANOVA analysis including sample type, tissue storage, and extraction kit as covariates prior to host taxonomic predictors. The authors also do mention these limitations on Lines 303 - 312 at the end of the discussion which I commend.

At a minimum I think the authors should clearly state these limitations strongly up front in the paper (abstract or introduction), primarily because the issues are more pronounced compared to other studies. This could be balanced by a clearer argument of the strengths of obtaining samples from wild organisms in natural habitats, which justifies some of the variation in approach. A more thorough discussion of likely effects of differences in specimen type (whole organism, stool, extracted gut, toe clipping...), storage medium, between kit variation, and lysis approach would help with details from the literature. The ideal scenario would be benchmarking differences between these approaches, but seems beyond the scope of this study.

Some key examples :

A. Samples within the same host class were of very different tissues. For instance some insects (e.g. cockroaches, honeybees) had guts extracted while others used the whole organism for DNA extraction (e.g. tsetse fly, hard tick). A more concerning example is amphibia where among four species sampled, one species extraction used a skin swab, one used dissected tadpole mouthparts, one used the whole organism, and one used a toe clipping. Birds and mammals tend to be more consistent using either dissected guts or fecal samples, and therefore relationships are likely more consistent with biology.

B. Tissue storage medium is known to bias microbiome profiles considerably as the authors note at line 309, and there was significant variation including: raw frozen samples, samples stored in varying concentrations of ethanol, and samples stored in RNALater.

C. As many species samples are extracted at different sites and with different kits, there is also likely critical variation in the lysing methods of samples. Whether bead beating (either in a proper bead beating machine or desktop vortexer), through chemical lysis, or mechanical lysis can play an enormous role on the detection of certain microbial partners, and while not noted these likely varied between different sites performing extractions.

I think this manuscript has significant value to the field, and this primary concern is not easily remedied without extensive field and lab work for identical sample processing. Therefore I think the best course would be to make this limitation very clear up front and discuss how individual factors may affect the analyses. Ultimately I think the strengths in bacteria to fungal comparison and use of wild animals outweigh the potential biases in sample handling, as long as the reader can clearly make this judgement for themselves.

Data Accessibility:

The SRA sequence archive is accessible and all samples appear clearly labeled by mycobiome and microbiome, and the github repository contains a well documented r-markdown script of downstream analyses. I was unable to locate the key metadata and dietary files, and could not locate scripts relating to the processing / quality control of raw reads, establishment into ASVs, and taxonomic assignment with DADA2. The r-markdown script appears to start with established community profiles, but scripts for processing raw sequences into profiles could not be located.

Additional points:

Figure 1A: Using inverse Simpson diversity is an appropriate alpha diversity measure, however it is likely not the best choice for the comparisons they are making and the display of this data. The issue is that the tables are rarefied to 500 reads per sample and with Simpson diversity this is likely driving the outlying values. A few individual samples have high diversity (index >35) while most are <5, leaving most species clustered at the bottom which makes between species variation difficult to examine. A better index to draw out the spread in diversity values between species would be Shannon's diversity index. Shannon also might be more appropriate for the low coverage and very different compositions of the microbiomes across species as it has a logarithmic transformation in the calculation which weights rarer organisms more heavily. Making distinguishing between species different more difficult is the overlay of individual samples with opaque white circles bordered in green or orange, as they frequently obscure the quartile and median black bars of the boxplots. Finally, the authors should clarify in the figure legend (like figure S1A: Median (\pm 25th and 75th percentiles)) what parts of the boxplot detail as some programs use different values (quartiles, mean versus median of central bar, that the circles are individual sample values).

Supplementary Methods: PCR amplification was done with 30 cycles which is somewhat high given the 10ng/uL input DNA concentration, which could increase PCR artifacts and off-target amplification. Maybe mention why 30 cycles was chosen.

Line 193-207, 225, 270-290: This network analysis is creative and informative. The repeated comparison that mammals have highly connected networks and insects and birds much less so,

however is the analytical result I would trust the least in this paper given the variation in sample collection and processing noted in supplementary table 1 (insects and birds have much higher variability than mammals). While this would be the biological pattern I would expect, I think the authors need to clearly note in discussion Line 280 that these patterns could also be influenced by sampling biases between host classes.

Line 256-269: This is an excellent discussion of phyllosymbiosis variation at different microbial taxonomic cutoffs.

Line 350: Samples were rarefied to 500 reads per sample for alpha diversity analyses, which is low coverage. The authors did verify that alpha diversity remained relatively stable when sampling at 1,000 or 2,500 reads (Figure S1). It appears that samples were rarefied a single time at each level, best practice would be to repeatedly rarefy at each sequence cutoff (e.g. rarefied 100 times and the mean alpha diversity values used across the rarefactions). I don't think they would see much variation in results however.

Line 378: The use of CLR and euclidian distance for beta diversity is an interesting choice, but likely meant to avoid loss of samples in rarefying cutoffs. Line 387: I think showing that results are similar with rarefaction should be presented in the supplement if it is being mentioned in the manuscript. The use of "(data not presented)" does not seem adequate here. If analyses are done with rarefaction on samples not CLR transformed I am wondering what distance metric was used (or were they CLR transformed after rarefying and then euclidian used)? It would be ideal to see an alternate metric like Jaccard distance or Bray-Curtis validated on the non-CLR transformed rarefied tables for the same PERMANOVA analysis in Table 2.

Summary: I found the analyses to be thorough, well executed, and proper to address the hypothesis in question. The primary weakness of the paper seems to result from sampling that was beyond the authors control for the most part, and is documented in detail in table S1. I think it should be more clearly stated early on as a potential weakness (given it cannot be easily remedied) so the reader can judge for themselves. Use of wild animals compared to lab / zoo animals is an enormous strength in my opinion, and the comparison of fungal and bacterial communities across so many host species is completely novel. The writing is clear and focused, and the discussion was thought provoking and robust. Overall this was quite an enjoyable and informative paper to read.

RSPB-2021-0552.R1 (Revision)

Review form: Reviewer 1

Recommendation

Accept with minor revision (please list in comments)

Scientific importance: Is the manuscript an original and important contribution to its field?

Good

General interest: Is the paper of sufficient general interest?

Good

Quality of the paper: Is the overall quality of the paper suitable?

Good

Is the length of the paper justified?

Yes

Should the paper be seen by a specialist statistical reviewer?

No

Do you have any concerns about statistical analyses in this paper? If so, please specify them explicitly in your report.

No

It is a condition of publication that authors make their supporting data, code and materials available - either as supplementary material or hosted in an external repository. Please rate, if applicable, the supporting data on the following criteria.

Is it accessible?

Yes

Is it clear?

Yes

Is it adequate?

Yes

Do you have any ethical concerns with this paper?

No

Comments to the Author

The authors have adequately addressed all of my comments and overall I think this is a nice paper worthy of publication in this journal.

I do have the following minor suggestions,

- I couldn't find the figure captions for the main figures, so found Fig 3D a bit hard to follow - however, if I understand correctly that the heatmaps represent the counts of positive interactions between different classes within phyla then it may be useful to represent this as a proportion of all potential interactions not an absolute count to allow for clearer comparison between hosts. Similarly, does the analysis of co-occurrence take account of the differing number of taxa between different host types / multiple hypothesis testing? May be good to make this clear either way.
- The positive correlation between actinobacteria-ascozymycota could equally be due to both sharing a very similar niche, and thus being in competition rather than cross-feeding one another - it might be useful to present both possibilities at line 270.
- Reading the abstract I wondered what you actually meant by "critical components" - being really pedantic I'd say just because fungi are present doesn't guarantee they're playing an important role in function for example. Personally I'd change or clarify this, but I recognise this is a matter of taste.

Review form: Reviewer 2 (Andrew Brooks)

Recommendation

Accept as is

Scientific importance: Is the manuscript an original and important contribution to its field?

Excellent

General interest: Is the paper of sufficient general interest?

Excellent

Quality of the paper: Is the overall quality of the paper suitable?

Good

Is the length of the paper justified?

Yes

Should the paper be seen by a specialist statistical reviewer?

No

Do you have any concerns about statistical analyses in this paper? If so, please specify them explicitly in your report.

No

It is a condition of publication that authors make their supporting data, code and materials available - either as supplementary material or hosted in an external repository. Please rate, if applicable, the supporting data on the following criteria.

Is it accessible?

Yes

Is it clear?

Yes

Is it adequate?

Yes

Do you have any ethical concerns with this paper?

No

Comments to the Author

This revision did a commendable job addressing the points raised. This is a very strong paper that adds to the body of work supporting phyllosymbiosis, and I'm sure the uniquely diverse dataset must have been quite difficult to assemble. It is especially heartening to see support in natural populations which have received less attention so far. I hope to see this in print in short order!

Decision letter (RSPB-2021-0552.R1)

09-Jul-2021

Dear Dr Meade

I am pleased to inform you that your manuscript RSPB-2021-0552.R1 entitled "Fungal microbiomes are determined by host phylogeny and exhibit widespread associations with the bacterial microbiome" has been accepted for publication in Proceedings B.

The referee(s) have recommended publication, but also suggest some minor revisions to your manuscript. Therefore, I invite you to respond to the referee(s)' comments and revise your manuscript. Because the schedule for publication is very tight, it is a condition of publication that

you submit the revised version of your manuscript within 7 days. If you do not think you will be able to meet this date please let us know.

[http://datadryad.org/submit?journalID=RSPB&manu=\(Document not available\)](http://datadryad.org/submit?journalID=RSPB&manu=(Document%20not%20available)) which will take you to your unique entry in the Dryad repository. If you have already submitted your data to dryad you can make any necessary revisions to your dataset by following the above link. Please see <https://royalsociety.org/journals/ethics-policies/data-sharing-mining/> for more details.

Sincerely,
Dr Maurine Neiman
Editor, Proceedings B
<mailto:proceedingsb@royalsociety.org>

Associate Editor:

Board Member: 1

Comments to Author:

Thank you for submitting a revised version of your manuscript. This has been assessed by the original reviewers. Both are happy with the revisions, and there are just a few remaining issues to fix (see detailed comments of Reviewer 2).

Reviewer(s)' Comments to Author:

Referee: 2

Comments to the Author(s)

This revision did a commendable job addressing the points raised. This is a very strong paper that adds to the body of work supporting phyllosymbiosis, and I'm sure the uniquely diverse dataset must have been quite difficult to assemble. It is especially heartening to see support in natural populations which have received less attention so far. I hope to see this in print in short order!

Referee: 1

Comments to the Author(s)

The authors have adequately addressed all of my comments and overall I think this is a nice paper worthy of publication in this journal.

I do have the following minor suggestions,

- I couldn't find the figure captions for the main figures, so found Fig 3D a bit hard to follow - however, if I understand correctly that the heatmaps represent the counts of positive interactions between different classes within phyla then it may be useful to represent this as a proportion of all potential interactions not an absolute count to allow for clearer comparison between hosts. Similarly, does the analysis of co-occurrence take account of the differing number of taxa between different host types / multiple hypothesis testing? May be good to make this clear either way.

- The positive correlation between actinobacteria-ascozymycota could equally be due to both sharing a very similar niche, and thus being in competition rather than cross-feeding one another
- it might be useful to present both possibilities at line 270.
- Reading the abstract I wondered what you actually meant by “critical components” - being really pedantic I’d say just because fungi are present doesn’t guarantee they’re playing an important role in function for example. Personally I’d change or clarify this, but I recognise this is a matter of taste.

Author's Response to Decision Letter for (RSPB-2021-0552.R1)

See Appendix A.

Decision letter (RSPB-2021-0552.R2)

14-Jul-2021

Dear Dr Meade

I am pleased to inform you that your manuscript entitled "Fungal microbiomes are determined by host phylogeny and exhibit widespread associations with the bacterial microbiome" has been accepted for publication in Proceedings B.

Data Accessibility section

Open Access

Paper charges

Sincerely,
Proceedings B
mailto:proceedingsb@royalsociety.org

Appendix A

Many thanks for the rapid assessment of our revised paper and the positive comments. We have now completed the additional revisions as requested below.

- I couldn't find the figure captions for the main figures, so found Fig 3D a bit hard to follow - however, if I understand correctly that the heatmaps represent the counts of positive interactions between different classes within phyla then it may be useful to represent this as a proportion of all potential interactions not an absolute count to allow for clearer comparison between hosts.

Figure 3 has now been updated to depict the frequency of interactions as proportions. We've also updated the figure legend to reflect this.

Similarly, does the analysis of co-occurrence take account of the differing number of taxa between different host types / multiple hypothesis testing? May be good to make this clear either way.

Yes, we corrected these permutation tests for multiple testing, and the co-occur package we used to detect the co-occurrences in the first place also corrects *p values*. Now clarified in the methods: "Here, we shuffled the fungal phylum data for our class-specific data of positive co-occurrences to estimate a null distribution of the expected frequency of co-occurrences., and corrected *p values* for multiple testing using False Discovery Rate with the 'p_adjust' function."

- The positive correlation between actinobacteria-ascozymycota could equally be due to both sharing a very similar niche, and thus being in competition rather than cross-feeding one another - it might be useful to present both possibilities at line 270.

Thanks for pointing this out. Now changed to:

"That we observe such a tight coupling between these two cross-kingdom taxonomic groups suggests they may be involved in cross-feeding metabolic networks within the avian gut, or alternatively may be in competition for similar niche space"

- Reading the abstract I wondered what you actually meant by "critical components" - being really pedantic I'd say just because fungi are present doesn't guarantee they're playing an important role in function for example. Personally I'd change or clarify this, but I recognise this is a matter of taste.

We have now removed this clause, so the sentence reads *"Using co-occurrence networks, we demonstrate marked variation across host taxonomy in patterns of covariation between bacterial and fungal abundances."*